# Ferroptosis-Inhibitory Difference between Chebulagic Acid and Chebulinic Acid Indicates Beneficial Role of HHDP

**DOI:** 10.3390/molecules26144300

**Published:** 2021-07-15

**Authors:** Lin Yang, Yangping Liu, Wenhui Zhang, Yujie Hua, Ban Chen, Quanzhou Wu, Dongfeng Chen, Shuqin Liu, Xican Li

**Affiliations:** 1School of Basic Medical Science, Guangzhou University of Chinese Medicine, Waihuan East Road No. 232, Guangzhou 510006, China; liny82416@gmail.com (L.Y.); cdf27212@21cn.com (D.C.); 2The Fourth Clinical Medical College, Guangzhou University of Chinese Medicine, Waihuan East Road No. 232, Guangzhou 510006, China; dryangpingliu@163.com; 3School of Chinese Herbal Medicine, Guangzhou University of Chinese Medicine, Waihuan East Road No. 232, Guangzhou 510006, China; 17630047507@163.com (W.Z.); huayujie0517@163.com (Y.H.); imchenban@foxmail.com (B.C.); gzywqz@gzucm.edu.cn (Q.W.); 20201110616@stu.gzucm.edu.cn (S.L.)

**Keywords:** ferroptosis-inhibition, ferrostatin-1, chebulagic acid, chebulinic acid, hydrolyzable tannin, HHDP

## Abstract

The search for a safe and effective inhibitor of ferroptosis, a recently described cell death pathway, has attracted increasing interest from scientists. Two hydrolyzable tannins, chebulagic acid and chebulinic acid, were selected for the study. Their optimized conformations were calculated using computational chemistry at the B3LYP-D3(BJ)/6-31G and B3LYP-D3(BJ)/6-311 + G(d,p) levels. The results suggested that (1) chebulagic acid presented a chair conformation, while chebulinic acid presented a skew-boat conformation; (2) the formation of chebulagic acid requires 762.1729 kcal/mol more molecular energy than chebulinic acid; and (3) the 3,6-HHDP (hexahydroxydiphenoyl) moiety was shown to be in an (*R*)- absolute stereoconfiguration. Subsequently, the ferroptosis inhibition of both tannins was determined using a erastin-treated bone marrow-derived mesenchymal stem cells (bmMSCs) model and compared to that of ferrostatin-1 (Fer-1). The relative inhibitory levels decreased in the following order: Fer-1 > chebulagic acid > chebulinic acid, as also revealed by the in vitro antioxidant assays. The UHPLC–ESI-Q-TOF-MS analysis suggested that, when treated with 16-(2-(14-carboxytetradecyl)-2-ethyl-4,4-dimethyl-3-oxazolidinyloxy free radicals, Fer-1 generated dimeric products, whereas the two acids did not. In conclusion, two hydrolyzable tannins, chebulagic acid and chebulinic acid, can act as natural ferroptosis inhibitors. Their ferroptosis inhibition is mediated by regular antioxidant pathways (ROS scavenging and iron chelation), rather than the redox-based catalytic recycling pathway exhibited by Fer-1. Through antioxidant pathways, the HHDP moiety in chebulagic acid enables ferroptosis-inhibitory action of hydrolyzable tannins.

## 1. Introduction

Ferroptosis is a recently discovered cell death pathway, which has been demonstrated to be driven by iron-dependent lipid peroxidation (LPD) [1,2,3]. The promotion of ferroptosis can kill cancer cells. Therefore, it may offer a novel therapeutic option for cancer, and it has accordingly become a key topic in the cell biology field [4,5,6,7,8].

In addition to the promotion of ferroptosis, its inhibition also represents a key topic because of its potential for rescuing pathologic cells. Ferroptosis inhibitors, thus, offer promise in the treatment of various noncancerous diseases, such as Parkinson’s disease, Alzheimer’s disease, atherosclerosis, and liver disease [9,10,11,12,13,14]. As such, some academic institutions have synthesized quite a few ferroptosis inhibitors, such as ferrostatin-1 (Fer-1) and liproxstatin-1 [15,16].

Recently, Perking University synthesized a novel ferroptosis inhibitor named PKUMDL-LC-101 [17]. Interestingly, it is similar to the UAMC-3203 inhibitor, synthesized by Antwerpen University in 2018 [18]. The similarity between PKUMDL-LC-101 and UAMC-3203 is reflected in the sulfanilamide moiety, which contains a secondary aromatic N-atom, as also found in Fer-1 and liproxstatin-1. Therefore, the above synthetic ferroptosis inhibitors can be classified as secondary aromatic amines, potentially leading to dangerous signals for the human body. In fact, secondary aromatic amine ferroptosis inhibitors reportedly have cytotoxic effects [15,19]. Furthermore, the toxic effect of sulfanilamide antibiotics has been well recognized since the 20th century [18,20,21].

Therefore, there is an urgent need to screen for safe and effective ferroptosis inhibitors from natural products. Previous studies have stated that flavonoid and monostilbenes could act as natural ferroptosis inhibitors [22,23,24,25], whereby the ferroptosis-inhibitory effect may arise from the presence of the phenolic –OH group.

Hydrolyzable tannins typically contain multiple phenolic –OH groups; accordingly, two hydrolyzable tannins, chebulagic acid and chebulinic acid, were selected for study (Figure 1). Coincidently, chebulagic acid and chebulinic acid typically coexist in the plant Terminalia chebula Retz [26], the dried fruit of which is named Kezi in TCM. Kezi is documented to be nontoxic and can be consumed as a medicinal congee [27].

To ensure their biological relevance, the two hydrolyzable tannins were studied using living bone marrow-derived mesenchymal stem cells (bmMSCs). The bmMSCs were prepared from rats and then treated with erastin to establish ferroptotic cell models [4,5]. The ferroptotic bmMSCs were further treated by the two acids and Fer-1, and the results were evaluated using a series of cellular methods. Lastly, the ferroptosis-inhibitory mechanisms underlying the activities of chebulagic acid, chebulinic acid, and Fer-1 were determined using experimental and theoretical chemical approaches.

## 2. Results and Discussion

Structurally, the HHDP moiety is the main characteristic used to distinguish chebulagic acid and chebulinic acid. Earlier studies typically ignored the stereochemistry of HHDP [28,29]. In fact, the covalent bridge in HHDP is a chiral axis. This is because, although the bridge is a σ-bond, the diester ring (rather than the glucopyranosyl ring) restrains its free rotation [30]; furthermore, the two aromatic rings are triggered at a dihedral angle. The chiral axis in HHDP may show different absolute configurations, (*R*)- or (*S*)-. Correspondingly, HHDP-containing tannins exist as two atropisomers [31]. From the perspective of organic stereochemistry, these two atropisomers are actually two diastereomers, some of which can be individually isolated from plants [32].

To explore the possible absolute configurations of the chiral axis in chebulagic acid, this study optimized its molecular shape using a quantum chemistry approach. The results indicated that HHDP showed an (*R*)- absolute configuration, agreeing with the previous literature [33]. This can be attributed to the fact that both the 6-ester group and the 3-ester group were attached to the ring, which presented a chair conformation (Figure 1B). In this configuration, the total energy of the molecule was 3612.8214 hartree.

For comparison, chebulinic acid was also calculated using the above approach. The results revealed that chebulinic acid possessed high single-point energy (*E* = −3614.0350 hartree) and a skew-boat conformation (Figure 1E,F). Accordingly, chebulagic acid always showed higher total energy than chebulinic acid, with a 762.1729 kcal/mol energy gap between the stabler chebulinic acid conformation and (*R*)-chebulagic acid. Despite chebulagic acid presenting a stable chair conformation, it was more unstable than chebulinic acid. Thus, stability could not be attributed to the presence of the (*R*)-HHDP moiety, which instead improved the reactivity of chebulagic acid.

Subsequently, both chebulagic acid and chebulinic acid were assayed using the H_2_DCFDA fluorescence method. As illustrated in Figure 2, both chebulagic acid and chebulinic acid displayed evident green fluorescence. Their intensities were weaker than that of the model (erastin) group, but stronger than that of the control group. These microscopy images suggested that both chebulagic acid and chebulinic acid could prevent LPO accumulation in ferroptotic bmMSCs treated by erastin. LPO accumulation has been documented to be a key characteristic of ferroptosis [24,34]. Thus, the inhibition of LPO accumulation implied that both chebulagic acid and chebulinic acid have ferroptosis-inhibitory potential.

To quantitatively determine the relative LPO inhibition level, flow cytometer analysis based on C11-BODIPY staining was introduced in the study, where Fer-1 (1 μM) was used as the positive control [35]. The positive control Fer-1 was observed to exhibit the highest inhibition level. At 5 μM concentration, chebulagic acid was superior to chebulinic acid (Figure 3A). 

LPO is known to be transformed from other ROS (such as ^•^O_2_^−^ [36] and ^•^OH [37]). Thus, the total ROS levels in cells were further measured using H_2_DCFDA-based flow cytometry. The results of H_2_DCFDA fluorescence determination suggested a decrease in the following order: Fer-1 < chebulagic acid < chebulinic acid (Figure 3B).

The inhibition of LPO and other ROS has been reported to improve cell viability [25]. Accordingly, the cell viability percentages increased in the following order Fer-1 > chebulagic acid > chebulinic acid (Figure 3C). Correspondingly, the cell death percentages were shown to decrease according to LDH release determination (Figure 3D) in the following order: Fer-1 < chebulagic acid < chebulinic acid. In general, 5 μM chebulagic acid could inhibit the ferroptosis of bmMSCs. Such a ferroptosis-inhibitory level was roughly equal to that exhibited by 1 μM Fer-1. 

The ferroptosis-inhibitory effect of Fer-1 is reportedly derived from its secondary aromatic amine N-atom. The N-atom mediates redox-based catalytic recycling, thereby exerting an inhibitory effect [16,20]. The recycling of catalysts certainly presents advantages of a low dose and high effectiveness.

As seen in Figure 1A, chebulagic acid contains no N-atom. Therefore, its dose was higher than that of Fer-1. However, its inhibitory effect at 50 μM was roughly equal to that of Trolox at 250 μM [38], indicating that chebulagic acid is an effective ferroptosis inhibitor. These observations, along with the previous studies, suggest that chebulagic acid can be used in stem cell transplantation therapy for some neurodegenerative diseases, such as Alzheimer’s disease [39,40].

To probe whether their ferroptosis inhibition is associated with an antioxidant action, the three inhibitors were investigated using PTIO^•^ inhibition, an in vitro antioxidant assay established by our team [41]. As seen in Appendix A and Table 1, the PTIO^•^ radical was efficiently inhibited by both tannins, suggesting their hydrogen atom transfer potential in the physiological solution [42,43]. Fer-1 had no available data regarding its insolubility in the physiological aqueous buffer (Table 1).

The IC_50_ value (in μM unit) was defined as the final concentration of 50% radical inhibition or relative reducing/chelating power, calculated by linear regression analysis and expressed as the mean ± SD (*n* = 3). The linear regression was analyzed by the Origin 2017 professional software. The IC_50_ values with different superscripts (a, b, or c) in the same row were significantly different (*p* < 0. 05). Trolox was the positive control (*, the positive control was sodium citrate instead of Trolox). The dose–response curves are listed in Appendix A. Ratio (1) was defined as IC_50_, _Trolox_/IC_50_, _chebulagic acid_. Ratio (2) was defined as IC_50_, _Trolox_/IC_50_,_chebulinic acid_. Ratio (3) was defined as IC_50_, _Trolox_/IC_50_, _Fer-1_. n.d., the data were not available for the insolubility. *The positive control was sodium citrate.

Subsequently, FRAP, ABTS^•+^ inhibition, and DPPH^•^ inhibition assays were also performed. As seen in Table 1 and Appendix A, chebulagic acid, chebulinic acid, and Fer-1 gave good dose–response curves for all three antioxidant assays (Table 1 and Appendix A). As seen in Table 1, the antioxidant levels (i.e., ratio values) of chebulagic acid, chebulinic acid, and Fer-1 differed, with values of 7.7, 6.4, and 1.7 Trolox equivalent, respectively. These data suggested that (*i*) chebulagic acid and chebulinic acid are very strong antioxidants, and (*ii*) the relative antioxidant levels of chebulagic acid and chebulinic acid correlate with their ferroptosis inhibition levels (Figure 2 and Figure 3). Therefore, on the basis of the above results and previous studies [22,23,24,25,44,45], the ferroptosis inhibition of both tannins was presumed to be via an antioxidant mechanism, attributed to the presence of multiple phenolic –OH groups [21,46,47]. It is worth mentioning that, the so-called “antioxidant mechanism” also includes SET-PT (single electron transfer–proton transfer) and SPLET (sequential proton loss electron transfer), apart from the above hydrogen atom transfer [48,49]. However, the net results of these mechanisms are generally identical with each other, i.e., the antioxidant molecule loses a hydrogen atom and the free radical obtains a hydrogen atom. Thus, the differences in antioxidant mechanisms do not hinder the further discussion in the study.

Unlike both tannins, Fer-1′s activity is reportedly via redox-based catalyst recycling [20]. Nevertheless, the interaction of Fer-1 with LPO is inevitable in ferroptotic cells. To explore the possible consequences of this interaction, Fer-1 was mixed with the 16-DOXYL -stearic acid free radical, a mimic of LPO [50,51]. UHPLC–ESI-Q-TOF-MS analysis revealed the production of a Fer-1/Fer-1 dimer (Figure 4), suggesting the possible effect of its interaction with LPO in ferroptotic cells. This would obviously result in increased metabolic risk, due to the potential cytotoxicity and genetic variation effects of this product [37,52]. In fact, such synthesized ferroptosis inhibitors led to a microgram-grade cytotoxic effect [15,19], and the metabolism risk of Fer-1 was recently revealed [18]. However, there have been no similar effects reported for chebulagic acid or chebulinic acid; thus, these tannin ferroptosis inhibitors can be considered safer than Fer-1.

The safety of both acids was further supported by the iron-binding pathway. As shown in Figure 5A, Fer-1′s UV-vis spectrum was adjusted upon being mixed with an iron(II) solution, implying the occurrence of iron(II) complexation. Iron binding was reported to involve the N-atom [20]. The N-atom, however, was found to possess a very low IP (ionization potential) value by our team, potentially rendering it incompatible in human cells [53].

Both tannin inhibitors greatly changed the UV/Vis spectrum in terms of strength and profile, as well as the solution color (Figure 5B). This implies their successful binding of iron. According to previous studies [54], iron binding may involve adjacent –OH groups, thereby forming seven-membered or five-membered rings (Figure 6). Such iron-binding reactions also occur in edible phytophenols and are, thus, regarded as safe [55,56,57].

Lastly, it should be noted that both tannins exhibited substantially different ferroptosis inhibition levels (Figure 2 and Figure 3) and ROS-scavenging levels (Table 1). This difference can could be attributed to the presence of 3,6-(*R*)-HHDP. As previously mentioned, the 3,6-(*R*)-HHDP moiety elevated the molecular energy by 762.1729 kcal/mol (Figure 1), thus improving reactivity and ferroptosis-inhibitory bioactivity. This finding will also benefit the understanding of other hydrolysable tannins, e.g., 1-*O*-galloyl-4,6-(*R*)-hexahydroxydiphenoyl (HHDP)-β-d-glucose [58], tellimagrandin I, and corilagin (Appendix A).

## 3. Materials and Methods

### 3.1. Chemicals, Biological Kits, Animals, and Software

Erastin (CAS number: 571203-78-6) was obtained from MedChemExpress (Monmouth Junction, NJ, USA). Ferrostatin-1 (CAS number: 347174-05-4) was purchased from Selleck Chemicals (Houston, TX, USA). Chebulagic acid (C_41_H_30_O_27_, CAS number: 23094-71-5, MW 954.664, purity 98%) and chebulinic acid (C_41_H_32_O_27_, CAS number: 18942-26-2, MW 956.68, purity 98%) were obtained from Chengdu Alfa Biotech. Ltd. (Chengdu, China). Furthermore, (±)-6-hydroxyl-2,5,7,8-tetramethylchromane-2-carboxylic acid (Trolox), 16-DOXYL-stearic acid free radical(CAS number: 53034-38-1), and 2,4,6-tripyridyl triazine (TPTZ) were obtained from Sigma-Aldrich (Shanghai, China). The 2-phenyl-4,4,5,5-tetramethylimidazoline-1-oxyl-3-oxide radical (PTIO^•^) was purchased from TCI Chemical Co. (Shanghai, China). 1,1-Diphenyl-2-picrylhydrazyl radical (DPPH^•^, C_18_H_12_N_5_O_6_) was obtained from Aladdin Chemical, Ltd. (Shanghai, China), while (NH_4_)_2_ABTS (2,2′-azino-*bis*(3-ethylbenzo-thiazoline-6-sulfonic acid diammonium salt)) was obtained from Amresco Chemical Co. (Solon, OH, USA). Water and methanol were of HPLC grade. FeCl_2_·4H_2_O and the other reagents of analytical grade were purchased from Guangdong Guanghua Chemical Plants Co., LTD (Shantou, China).

The complete medium with glucose for SD rat bone marrow mesenchymal stem cells was purchased from Cyagen Biosciences (Santa Clara, CA, USA); fetal bovine serum (FBS) and trypsin were obtained from Molecular Probes (Carlsbad, CA, USA). The C11-BODIPY probe was purchased from Molecular Probes (Carlsbad, CA, USA). The annexin V/propidium iodide (PI) determination kit was purchased from BD Biosciences (Franklin Lakes, NJ, USA). Cell Counting Kit-8 kit was purchased from the Dojindo Chemistry Research Institute (Kumamoto, Japan).

Sprague-Dawley rats (4 weeks) were obtained from the Animal Center of the Guangzhou University of Chinese Medicine. Procurement, maintenance, and treatment of the animals were performed under the supervision of the Institutional Animal Ethics Committee of the Guangzhou University of Chinese Medicine. The Gaussian 16 C.01 program was purchased from Guangzhou Molcalx Ltd. (Guangzhou, China).

### 3.2. Conformation Optimization and Single-Point Energy Calculations Based on Computational Chemistry

The conformations of chebulinic acid and chebulagic acid were optimized at the B3LYP-D3(BJ)/6-31G level with no imaginary frequency, and the single-point energies of the optimized geometries were determined at the B3LYP-D3(BJ)/6-311+G(d,p) level. The calculations were performed using the Gaussian 16 C.01 program, while the optimized structures were visualized using the VMD program.

### 3.3. Characterization of Mitochondrial ROS in Ferroptotic bmMSCs

Bone marrow was harvested from the femur and tibia of rats based on our previous method [54]. Briefly, the basal medium containing 10% FBS was used to dilute the marrow. The diluted marrow was then centrifugated at 900× *g* for 30 min on 1.073 g/mL Percoll to prepare bmMSCs. The prepared bmMSCs were then detached with 0.25% trypsin and were further cultured in flasks at 1 × 10^4^ cells/cm^2^ [3,59,60].

The above bmMSCs were seeded at 1 × 10^5^ cells/well into 12-well plates and then cultured for 24 h. The cultured bmMSCs were classified into four groups, i.e., control, model, positive control, and sample. The bmMSCs in the control group were incubated for 12 h in the basal medium. Those in the model group, positive control, and sample groups, however, were incubated with erastin (10 μM) to create ferroptotic damage. Then, the mixture of erastin and medium was removed. The model group, however, was further incubated for 12 h in the above basal medium without sample solution, while the sample group was incubated for 12 h in the above basal medium with sample solution. The sample solutions, however, were at different concentrations. Lastly, the positive control was incubated for 12 h in the same medium, with 1.0 μM Fer-1.

To characterize the mitochondrial ROS concentration, the bmMSCs in the above four groups were incubated with 4′,6-diamidino-2-phenylindole (DAPI) staining. Followed by a 30 min incubation period, mitochondrial ROS were assayed using the H_2_DCFDA probe (5 μM) and a mitochondrial fluorescent probe [61].

### 3.4. Flow Cytometry Using the H2DCFDA Probe to Determine the Total Intracellular ROS Concentration

The bmMSCs in the above four groups were used to determine the total intracellular ROS concentration using flow cytometry and the H2DCFDA probe. Briefly, bmMSCs were rinsed using PBS two times, and subsequently interacted with 5 μM H_2_DCFDA staining solution in PBS at 37 °C for 30 min. The stained bmMSCs were collected with 0.05% trypsin solution and then suspended in fresh medium. The supernate was instantly determined by means of a flow cytometer.

### 3.5. Flow Cytometry Using the C11-BODIPY Probe to Assess LPO Accumulation

The prepared bmMSCs in Section 3.3 were assessed for their LPO accumulation [59,62]. In brief, the bmMSCs were seeded at 1 × 10^6^ cells/well into 12-well plates. After 24 h culture, bmMSCs were classified into control group, erastin group, erastin + Fer-1 group, and sample group, as mentioned by the literature [63]. In the control group, the bmMSCs were incubated for 12 h in stem cell basal medium. In the erastin and sample groups, bmMSCs were incubated with the mixture of 10 μM erastin and sample at different concentrations. In the erastin + Fer-1 group, bmMSCs were incubated with the mixture of 10 μM erastin and 1 μM Fer-1.

After incubation for 12 h, the mixture of erastin and medium was discarded. The erastin and erastin + Fer-1 groups were incubated for 12 h in the above basal medium, while the sample groups were incubated for 12 h in the above basal medium with sample. Following incubation, the cells were rinsed with PBS two times. The adhered cells were interacted with 2.5 µM C11-BODIPY staining solution in PBS and then harvested by means of 0.05% trypsin solution. Subsequently, the supernate was taken out from the fresh medium and immediately assessed using a flow cytometer.

### 3.6. Lactate Dehydrogenase (LDH) and CCK-8 Determinations

The LDH and CCK-8 determinations were conducted using the method described by Wenzel [64]. The method utilized a 96-well plate to seed bmMSCs at 1 × 10^4^ cells/well. These bmMSCs were subsequently classified into the aforementioned four groups, and then were treated by the method of Section 3.3.

However, the incubation period was 11 h for the LDH determination. Then, the LDH release reagent was added with the maximum enzyme activity to each group for 1 h. Subsequently, the plate was centrifuged for 5 min at 400× *g*. A volume of 120 μL was removed from each well and added to another blank 96-well plate, respectively. The LDH detection solution (60 μL) was added into each well, before incubating the plate for an additional 30 min. The A_490 nm_ was determined on a Bio-Kinetics reader (PE-1420; Bio-Kinetics Corporation, Sioux Center, IA, USA). For the CCK-8 determination, the cells incubated for 11 h were treated with 90 μL of RASMX-90011 and 10 μL of CCK-8 solution for 2 h. The A_450 nm_ value was determined on a Bio-Kinetics reader.

### 3.7. PTIO^•^ Inhibition Determination

The PTIO^•^ inhibition determination was conducted according to our previously published method [42]. Briefly, the PTIO^•^ radical was dissolved in pH 7.4 phosphate buffer to prepare a PTIO^•^ solution; the samples were prepared using methanol (1 mg/mL). Samples at various volumes (*x* = 2–10 μL) were mixed with phosphate buffer at pH 7.4 (20 − *x* μL) and treated with PTIO^•^ solution (80 μL). After incubation at 37 °C for 50 min, the product mixture was analyzed by measuring the absorbance at 560 nm on a microplate reader (Multiskan FC, Thermo Scientific, Shanghai, China). The PTIO^•^ inhibition percentage was calculated as follows:$$\mathrm{Inhibition}\;(\%)=\frac{A_{0}-A}{A_{0}}\times 100$$
where *A*_0_ is the absorbance at 560 nm of the control sample (without test agent) and *A* is the absorbance at 560 nm of the reaction mixture (with sample).

### 3.8. Fe^3+^-Reducing Antioxidant Power (FRAP) Determination

The FRAP determination was carried out as per the method described by Benzie and Strain [65,66], which was slightly modified in our previous studies [67,68]. Briefly, the FRAP reagent was freshly prepared by mixing 10 mM TPTZ, 20 mM FeCl_3_, and 0.25 M acetate buffer (pH 3.6) at a ratio of 1:1:10. Samples (0.2 mg/mL, *x* = 2–10 μL) were added to methanol (20 − *x* µL) and treated with 80 µL of FRAP reagent. After incubation for 30 min, the absorbance of the mixture was measured at 593 nm (A_593 nm_) on a microplate reader (Multiskan FC, Thermo Scientific, Shanghai, China). The relative reducing antioxidant power of the sample as compared to the maximum absorbance was calculated using the following formula:$$\mathrm{Relative}\;\mathrm{reducing}\;\mathrm{power}\;\left(\%\right)=\frac{A-A_{\min}}{A_{\max}-A_{\min}}\times 100$$
where A_min_ is the lowest *A*_593 nm_ value recorded in the experiment, *A* is the *A*_593 nm_ value of the Table A_593nm_ value recorded in the experiment.

### 3.9. DPPH^•^ Inhibition and ABTS^+•^ Inhibition Determination

The DPPH^•^ antioxidant activity was determined as previously described [69]. Briefly, 80 μL of a DPPH^•^ methanolic solution (0.1 mol/L) was mixed with sample methanolic solutions (*x* = 2–10 μL, 0.05 mg/mL) and methanol (20 − *x* μL). The mixture was maintained at room temperature for 6 min, and the absorbance was measured at 519 nm on a microplate reader. After a similar setup, the absorbance was measured at 734 nm on a microplate reader for the ABTS^+•^ inhibition assay. The percentage of DPPH^•^ inhibition activity and percentage of ABTS^+•^ inhibition activity were calculated using the formula presented above for the PTIO^•^ inhibition assay, where A_0_ is the absorbance of the control sample, and A is the absorbance of the reaction mixture.

### 3.10. UHPLC–ESI-Q-TOF-MS Analysis of 16-DOXYL-stearic Acid Free Radical Reaction Products with Ellagitannins

The reaction of 16-DOXYL-stearic acid^•^ with 4.0 mg/mL chebulagic acid proceeded under the conditions described in a previous method [70]. Briefly, a methanol solution of the sample was mixed with a methanol/6-DOXYL-stearic acid^•^ solution with a molar ratio of 1:2, and the resulting mixture was incubated for 24 h at room temperature in the dark. Subsequently, the product mixture was diluted 50-fold using methanol and passed through a 0.22 μM filter for UHPLC–ESI-Q-TOF-MS analysis [71].

The UHPLC–ESI-Q-TOF-MS analysis was based on a Q-TOF-MS system with chromatographic column. The Q-TOF-MS system was Triple TOF 5600*^plus^* mass spectrometer (AB SCIEX, Framingham, MA, USA); the chromatographic column was Phenomenex Luna C_18_ column (2.1 mm i.d. × 100 mm, 1.6 μM, Phenomenex Inc., Torrance, CA, USA).

In the present study, the mobile phase through the chromatographic column was designed as a mixture of methanol (phase A) and 0.1% formic acid water (phase B). The mobile phase flow rate was at 0.2 mL/min, and changed by the following gradient elution program: 0–2 min, 30% B; 2–10 min, 30→0% B; 10–12 min, 0→30% B. The sample injection volume was 3 μL. The Q-TOF-MS analysis was equipped with an ESI source, which was run in negative ionization mode from 100 to 2000 Da. The system was run with the following parameters: ion spray voltage, −4500 V; ion source heater temperature, 550 °C; curtain gas pressure (CUR, N_2_), 30 psi; nebulizing gas pressure (GS_1_, air), 50 psi; Tis gas pressure (GS_2_, air), 50 psi. The declustering potential (DP) was set at −100 V, whereas the collision energy (CE) was set at −45 V with a collision energy spread (CES) of 15 V.

The above experiments were repeated using chebulinic acid and Fer-1. The concentration of Fer-1 was 4.28 mg/mL.

### 3.11. Statistical Analysis

The data were recorded as the mean values of triplicate analyses. The dose–response curves were plotted using Origin 2017 professional software (OriginLab, Northampton, MA, USA). The IC_50_ value characterized the final concentration, achieving 50% PTIO^•^ radical inhibition (or DPPH^•^ radical inhibition, ABTS^•+^ radical inhibition, Fe^3+^-reducing relative reducing power) [72]. Statistical comparisons were conducted using one-way analysis of variance (ANOVA) to detect significant differences using SPSS 13.0 software (SPSS Inc., Chicago, IL, USA) for Windows.

## 4. Conclusions

Two hydrolyzable tannins, chebulagic acid and chebulinic acid, can act as natural ferroptosis inhibitors. Their ferroptosis inhibition is mediated by regular antioxidant pathways (ROS scavenging and iron chelation), rather than the redox-based catalytic recycling pathway exhibited by Fer-1. Both acids can be considered safer than Fer-1 with respect to their inhibitory pathways and inhibition products. The HHDP moiety in chebulagic acid can, however, enable antioxidant pathways to elevate the ferroptosis-inhibitory level. The findings will benefit other hydrolyzable tannins bearing HHDP moiety.

## Figures and Tables

**Figure 1 molecules-26-04300-f001:**
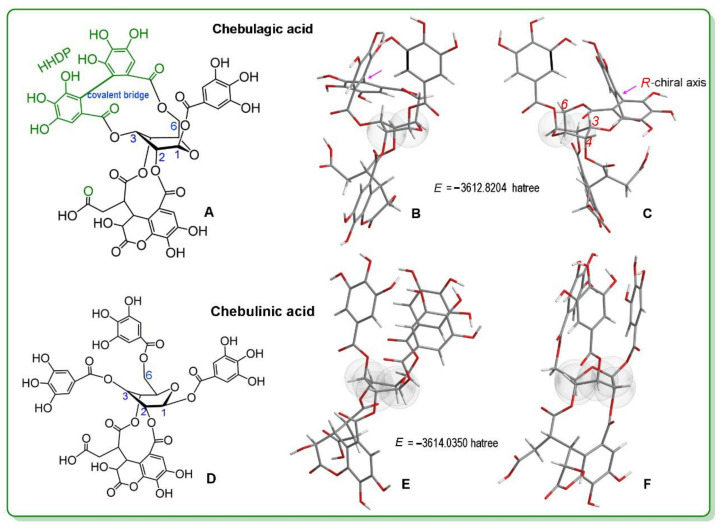
Structures, stable conformations, and molecular single point energies (*E*) of chebulagic acid and chebulinic acid. (**A**), structure of chebulagic acid; (**B**), stable conformation of chebulagic acid; (**C**), back view of (**B**); (**D**), structure of chebulinic acid; (**E**), stable conformation of chebulinic acid; (**F**), left view of E. HHDP, hexahydroxydiphenoyl; *E*, single point energy. Δ*E* = $E_{\mathrm{chebulinic}\;\mathrm{acid}}-E_{\mathrm{chebulagic}\;\mathrm{acid}}$ = 1.2146 Hatree = 762.1729 kcal/mol. The dot cloud indicates the glucopyranosyl ring. The pink arrows in B and C indicate the covalent bridge. *R*, absolute stereoconfiguration of the chiral axis).

**Figure 2 molecules-26-04300-f002:**
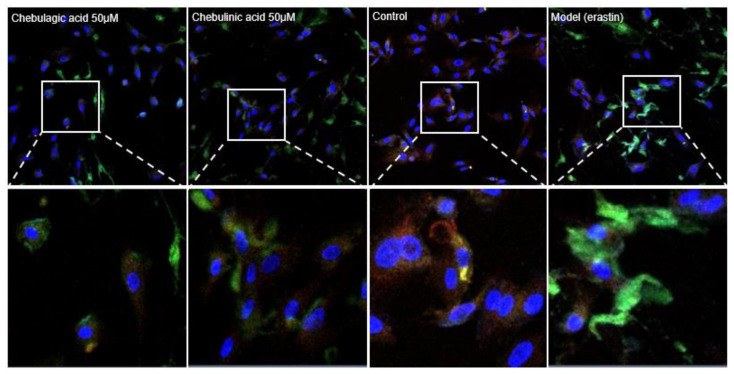
Microscopy images of fluorescent staining. Chebulagic acid 50 μM group, bmMSCs treated with erastin + 50 μM chebulagic acid; chebulinic acid 50 μM group, bmMSCs treated with erastin + 50 μM chebulinic acid; control group, bmMSCs without erastin treatment; model group, erastin-treated bmMSCs. The green fluorescence resulted from H2DCFDA dye and indicates the reactive oxygen species (ROS) accumulation; the blue fluorescence resulted from DAPI (4’,6-diamidino-2-phenylindole) dye and indicates the cell nuclei; the red–blue fluorescence resulted from BBcell ProbeTM dye and indicates the mitochondria.

**Figure 3 molecules-26-04300-f003:**
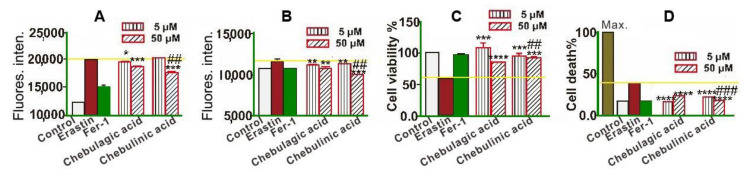
(**A**) BODIPY-probed flow cytometry analysis; (**B**) H2DCFDA-probed flow cytometry analysis; (**C**) cell counting kit-8 (CCK-8) analysis; (**D**) lactate dehydrogenase (LDH) release analysis. Control group, bmMSCs without erastin treatment; erastin group, 10 μM erastin-treated bmMSCs; Fer-1 group, bmMSCs treated with 10 μM erastin + 1 μM Fer-1; chebulagic acid group, bmMSCs treated with 10 μM erastin + chebulagic acid; chebulinic acid group, bmMSCs treated with 10 μM erastin + chebulinic acid. Data are presented as mean ± SD from three independent experiments. * *p* < 0.05, ** *p* < 0.01, *** *p* < 0.001, **** *p* < 0.0001 compared with the model group; ## *p* < 0.01, ### *p* < 0.001 compared with the chebulagic acid group at the same concentration.

**Figure 4 molecules-26-04300-f004:**
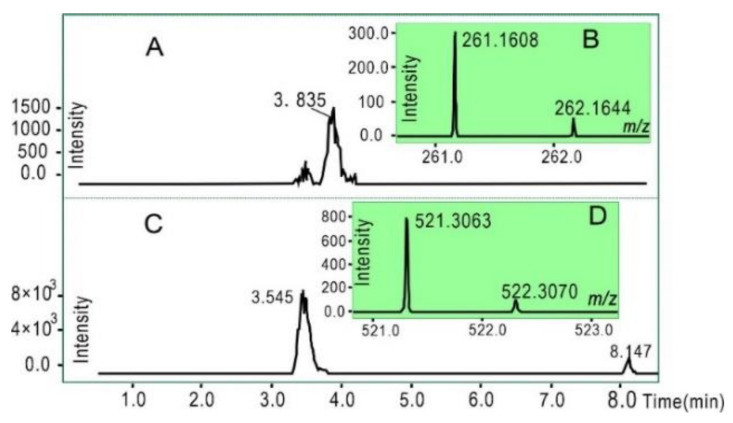
UHPLC–ESI-Q-TOF-MS analysis of Fer-1 with 16-DOXYL-stearic acid free radical: (**A**) chromatographic profile of standard Fer-1; (**B**) MS spectrum of standard Fer-1 (Rt = 3.835 min); (**C**) MS spectrum of Fer-1 dimer; (**D**) MS spectrum of Fer-1 dimer (Rt = 3.545 min). The ESI was run in negative ionization mode.

**Figure 5 molecules-26-04300-f005:**
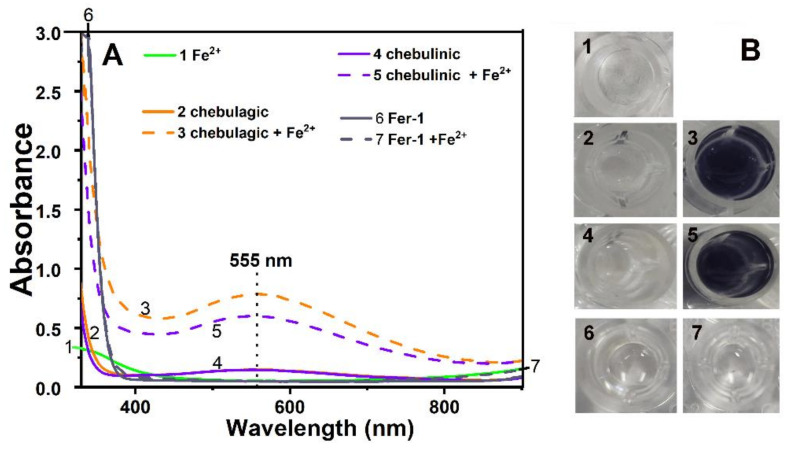
(**A**) Visible light spectra of high concentrations of chebulagic acid, chebulagic acid/iron(II), chebulinic acid, and chebulinic acid/iron(II); (**B**) colors of the various solutions. 1, 50.3 mM iron (II); 2, 0.3 mM chebulagic acid; 3, 0.3 mM chebulagic acid + 50.3 mM iron(II); 4, 0.3 mM chebulinic acid; 5, 0.3 mM chebulinic acid + 50.3 mM iron(II); 6, 0.3 mM Fer-1; 7, 0.3 mM Fer-1 + 50.3 mM iron(II).

**Figure 6 molecules-26-04300-f006:**
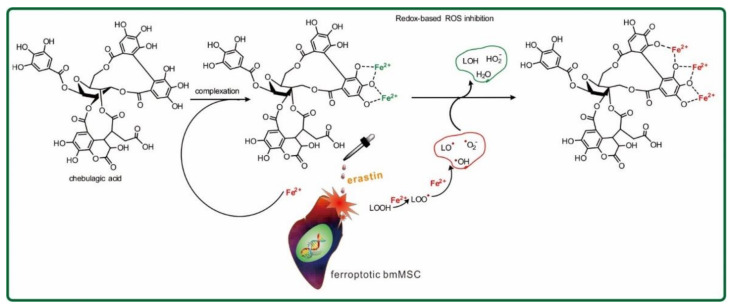
Illustration of antioxidant mechanisms underlying ferroptosis inhibition by chebulagic acid.

**Table 1 molecules-26-04300-t001:** IC_50_ values (μM) of chebulagic acid, chebulinic acid, Trolox, and Fer-1 in antioxidant determinations.

Determinations	Chebulagic Acid	Chebulinic Acid	Trolox	Fer-1	Ratio Values		
					(1)	(2)	(3)
PTIO^•^ inhibition	40.1 ± 1.5	56.4 ± 5.0	310.3 ± 14.8	*n*.d.	7.7	5.5	*n*.d.
FRAP	12.8 ± 1.1	22.5 ± 2.3	48.7 ± 2.5	36.3 ± 1.5	3.8	2.2	1.3
ABTS^•^ ^+^ inhibition	3.4 ± 0.0	3.5 ± 0.1	35.7 ± 0.8	14.9 ± 0.8	10.5	10.2	2.4
DPPH^•^ inhibition	4.0 ± 0.1	4.5 ± 0.4	34.9 ± 4.8	23.5 ± 0.7	8.7	7.8	1.5

*n*.d. not determined.

## Data Availability

Data are contained within the article or Appendix A. The data presented in this study are available in this manuscript.

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
