# Peer review of "Ferroptosis-Inhibitory Difference between Chebulagic Acid and Chebulinic Acid Indicates Beneficial Role of HHDP"

_molecules, 2021, doi:10.3390/molecules26144300_

Round 1

Reviewer 1 Report

This manuscript described the ferroptosis inhibitor properties of chebulagic acid and chebulinic acid and the important of HHDP group. Authors used computer software to calculated the stable conformation of HHDP group and confirm it is R conformation. However, the atropisomerism at the HHDP group in chebulagic acid is R conformation which already be reported in Chem. Pharm. Bull., 28, 3713 (1980). And thus authors should draw the structure of chebulagic acid as this reference.

In P2, L.40 and L.43, point energies (E), the E should italic as in figure 1.

In P2, L45, S absolute …, S should be R.

In Figure 2, the magnify square should be same size for compare the different purpose.

In Figure 3, the figure legend describe were differ as figure used. Such as Erastin is model group, positive control group is Fer-1 in figure legend. The significant differences marks ###, *** were compare with what?  Why the cell death % of chebulagic acid at 50 uM is higher than 5 uM? It showed toxicity but in P.4, L137-L138 the described this concentration can complete inhibition the ferropotosis? The concentration of erastin should be given.

In Figure 4, ESI was in negative mode should be mention. Figure 4A also can found the same retention time (3.545 min) peak as Figure 4C. Is it the same dimer of Fer-1?

In Figure 5, the figure legend does not complete describe the 10 and 11 means.

Figure 7 should be Figure 6.

 P.7, L.214-L.215, (S)-HHDP should be (R)-HHDP. Thus in suppl.2 the structures all has (S)-HHDP, the molecular energy difference is the same or not? Authors should use other example for (R)-HHDP.

P.7, L230 and P.10, L353, L355, L357, The correct name is 16NS, 2-(14-Carboxytetradecyl)-2-ethyl-4,4-dimethyl-3-oxazolidinyloxy (16-DOXYL-stearic acid).

P.8, L284, Section 3.2 should be section 3.3.

P8, L.286, 106 cells/well should be 10 superscript 6 cells/well.

P8, L287, L, 290, ferrostatin should be ferrostatin-1 or Fer-1.

P9, L327, the equation: Inhibition %=  ….x100%      should be Inhibition (%)=….x100

P10, L340, same as above.

P10, L349-352, Only mention the 519nm for DPPH. How about ABTS?

P10, L355, what is the concentration of chebulagic acid and chebulinic acid?

P10, L373, “Tis” gas pressure?

Reviewer 2 Report

Comments of the manuscript molecules-1275082-V1

Entitled: Ferroptosis-Inhibitory Difference between Chebulagic Acid and Chebulinic Acid Indicates Beneficial Role of HHDP

This work reports on ferroptosis inhibition by two hydrolysable tannins. The abstract well reflects to the content of the manuscript. The introduction clearly presents the challenges of finding inhibitors of ferroptosis. The manuscript is well structured, with high quality figures (except figure 5) and respect scientific formats. The manuscript is easily read.

In results, comparison of chebulagic acid and chebulinic acid using computational chemistry tools is fine.

Redox properties of chebulagic acid and chebulinic acid are badly analyzed, it should be analyzed by computational chemistry tools also. Radical scavenging properties need to study reactivity and thermodynamic of three chemical pathways: HAT (H-atom Transfer), SET-PT (Single electron transfer-proton transfer), and SPLET (Sequential proton loss electron transfer). Methodology applied by The et al 2019 (DOI: 10.1155/2019/4360175) for isoflavones and Murakami et al 2015, (in vivo 29: 341-350) for stilbenes, should be employed for estimating Redox properties of chebulagic acid and chebulinic acid. If authors consider it is not in the objectives of their paper, they could mention in discussion this future perspective.

The H2DCFDA assay provides ROS accumulation within living cells. It is misleading to consider that it “indicates the lipid peroxidation (LPO) accumulation” (line 121 page 4). This sentence should be corrected.

Comparison among the four antioxidant assays should be more explicitly explained (line 167 page 5). PTIO* assay test the hydrogen-atom transfer potential. FRAP corresponds to Ferric reducing antioxidant power. ATBS*+ assay can be quenched by both electron (fast kinetic) and hydrogen atom transfer (slow kinetic). DPPH* assay measures the capacity of an antioxidant to reduce the chemical radical DPPH° by hydrogen transfer. Is it relevant to give an average ratio score of four antioxidant assays that have different reactional mechanisms? I suggest to remove this average score.

Three curves of figure 5B are not identified. Figure 5 need to be refine.

Material and method part is sufficiently developed and well described.

This is a valuable manuscript. The results are sufficiently rare and of definite scientific interest to be published after a MINOR REVISION.

Reviewer 3 Report

The manuscript is of high quality. The authors study in details the ferroptosis inhibition by Chebulagic Acid and Chebulinic Acid. All assays used are robust, with important results on the structural differentiations effects on the activity. The authors not only perform the bioassays but also support them with high-resolution mass spectrometry to strengthen their findings. In this content, i suggest acceptance in its current form.  

Round 2

Reviewer 1 Report

This manuscript has been revised mostly.